# Targeted Metabolomics Identifies Plasma Biomarkers in Mice with Metabolically Heterogeneous Melanoma Xenografts

**DOI:** 10.3390/cancers13030434

**Published:** 2021-01-23

**Authors:** Daniela D. Weber, Maheshwor Thapa, Sepideh Aminzadeh-Gohari, Anna-Sophia Redtenbacher, Luca Catalano, René G. Feichtinger, Peter Koelblinger, Guido Dallmann, Michael Emberger, Barbara Kofler, Roland Lang

**Affiliations:** 1Research Program for Receptor Biochemistry and Tumor Metabolism, Department of Pediatrics, University Hospital of the Paracelsus Medical University, 5020 Salzburg, Austria; d.weber@salk.at (D.D.W.); s.aminzadeh-gohari@salk.at (S.A.-G.); anna.redtenbacher@stud.pmu.ac.at (A.-S.R.); l.catalano@salk.at (L.C.); r.feichtinger@salk.at (R.G.F.); 2BIOCRATES Life Sciences AG, 6020 Innsbruck, Austria; maheshwor.thapa@biocrates.com (M.T.); guido.dallmann@biocrates.com (G.D.); 3Department of Dermatology and Allergology, University Hospital of the Paracelsus Medical University, 5020 Salzburg, Austria; p.koelblinger@salk.at; 4Patholab Salzburg, 5020 Salzburg, Austria; emberger@pathologie-salzburg.at

**Keywords:** melanoma, targeted metabolomics, cancer metabolism, Warburg effect, lipid metabolism, beta-alanine metabolism, metabolic biomarker

## Abstract

**Simple Summary:**

Studying the metabolome, the complete set of metabolites found in a biological sample, helps to understand how metabolism differs between healthy and diseased individuals. Malignant skin cancers (melanomas) differ greatly in their mutational landscapes and metabolism among individuals. Understanding these differences is important for accurate treatment of melanoma patients. The aim of our study was to characterize the metabolomes of genetically different human melanoma cell lines transplanted into mice. We found that melanomas, regardless of mutation status, differed greatly in their lipid profiles, which are important metabolites for energy production. Moreover, we identified seven metabolites with the potential to distinguish healthy from melanoma-bearing mice. We hope that the metabolomic differences detected in the mouse model may be reproducible in humans and potentially lead to metabolism-based therapeutic approaches for melanoma patients.

**Abstract:**

Melanomas are genetically and metabolically heterogeneous, which influences therapeutic efficacy and contributes to the development of treatment resistance in patients with metastatic disease. Metabolite phenotyping helps to better understand complex metabolic diseases, such as melanoma, and facilitates the development of novel therapies. Our aim was to characterize the tumor and plasma metabolomes of mice bearing genetically different melanoma xenografts. We engrafted the human melanoma cell lines A375 (BRAF mutant), WM47 (BRAF mutant), WM3000 (NRAS mutant), and WM3311 (BRAF, NRAS, NF1 triple-wildtype) and performed a broad-spectrum targeted metabolomics analysis of tumor and plasma samples obtained from melanoma-bearing mice as well as plasma samples from healthy control mice. Differences in ceramide and phosphatidylcholine species were observed between melanoma subtypes irrespective of the genetic driver mutation. Furthermore, beta-alanine metabolism differed between melanoma subtypes and was significantly enriched in plasma from melanoma-bearing mice compared to healthy mice. Moreover, we identified beta-alanine, *p*-cresol sulfate, sarcosine, tiglylcarnitine, two dihexosylceramides, and one phosphatidylcholine as potential melanoma biomarkers in plasma. The present data reflect the metabolic heterogeneity of melanomas but also suggest a diagnostic biomarker signature for melanoma screening.

## 1. Introduction

Malignant melanoma is the leading cause of skin cancer-related death, even though it accounts for only 1% of skin cancers, and its incidence is increasing [1]. Cutaneous melanoma has been categorized into four major genomic subtypes: v-Raf murine sarcoma viral oncogene homolog B (BRAF) mutant (52%), neuroblastoma RAS viral oncogene homolog (NRAS) mutant (28%), neurofibromin 1 (NF1) (14%) mutant, and triple-wildtype (6%), the latter characterized by a lack of BRAF, NRAS, or NF1 mutations but enrichment of tyrosine kinase (KIT) mutations [2]. Each of the identified driver mutations contributes to the constitutive activation of the mitogen-activated protein kinase (MAPK) and/or phosphoinositide 3-kinase (PI3K)-signaling pathways, leading to uncontrolled cell growth and making the tumors susceptible to targeted therapies such as BRAF- or mitogen-activated protein kinase kinase (MEK)-inhibitors [2].

Besides genetic heterogeneity, melanoma cells present a variety of metabolic phenotypes that depend on both intrinsic oncogenic pathways and extrinsic factors in the tumor microenvironment (TME) [3,4,5]. The most recognized metabolic alteration of cancer cells is their utilization of glucose as the predominant substrate for energy production, inducing a shift from oxidative phosphorylation (OXPHOS) to glycolysis even if oxygen is present (Warburg effect) [6,7]. Metabolic flux profiling of a panel of melanoma cell lines and melanocytes revealed an elevated glycolytic rate in melanoma cells compared to melanocytes, but also functional tricarboxylic acid (TCA) cycle flux even under hypoxia, suggesting that energy production in melanoma cells is not exclusively glycolytic [8]. Moreover, we and others have shown that human melanomas exhibit a variable but distinct metabolic signature, as some patient-derived melanomas and melanoma cell lines expressed the classic Warburg phenotype, whereas others maintained significant, high levels of OXPHOS [9,10,11,12,13]. One explanation for the high OXPHOS activity of some melanomas might be the so-called “reverse Warburg effect”. According to this model, tumor cell surrounding cancer associated fibroblasts (CAFs), which highly depend on glycolysis due to mitochondrial inactivity, secrete lactate via the monocarboxylate transporter (MCT) 4 into the TME. In turn, tumor cells take up lactate via MCT1 from the TME to fuel OXPHOS [14]. A similar concept called “metabolic symbiosis” has been described, which involves hypoxic glycolytic tumor cells secreting lactate into the TME and thus providing oxidative tumor cells with lactate for energy production [15]. Interestingly, elevated expression of MCT1 and MCT4 has been correlated with melanoma progression [16].

In melanoma, MAPK and PI3K pathway activation induces glycolysis and its decoupling from the TCA cycle via stimulation of hypoxia-inducible factor 1α (HIF1α) and the v-MYC avian myelocytomatosis viral oncogene homolog (MYC) [17,18]. Moreover, in glycolytic melanomas, HIF1α suppresses transcription of the microphthalmia-associated transcription factor (MITF), which regulates peroxisome proliferator-activated receptor γ 1-α (PGC1α), resulting in OXPHOS inhibition [12]. However, in high-OXPHOS melanomas, MITF is not suppressed by HIF1α but is promoted by the mammalian target of rapamycin (mTOR), leading to the transcription of PGC1α, a transcriptional coactivator which induces the expression of different OXPHOS genes [11]. PGC1α expression levels have been associated with the metabolic state of melanomas and the response to MEK inhibition independent of BRAF or NRAS mutation status [11,19].

Besides glucose and pyruvate, glutamine is an important anaplerotic substrate to feed the TCA cycle. Melanoma cells are able to switch from glucose to glutamine as an alternative carbon source, which decouples mitochondrial TCA cycle activity from cytosolic glycolysis, highlighting a robust and flexible metabolic phenotype to cope with physiological challenges such as hypoxia [8,20]. Moreover, glutamine can support fatty acid (FA) synthesis in melanoma cells by running the TCA cycle in the reverse direction [20]. Dysregulation of lipid metabolism, including various alterations in FA metabolism, has been associated with melanoma growth and aggressiveness [21]. Fatty acid synthase (FASN), the rate limiting enzyme in FA synthesis, is upregulated in many cancer types, including melanoma, due to constitutive activation of the transcription factor sterol regulatory element-binding protein 1c (SREBP1c) driven by the MAPK and PI3K/AKT pathways [22,23,24,25]. Rapidly proliferating cancer cells require large amounts of FAs in order to build new cell membranes, the structural foundation of cells and organelles. Thus, increased FASN activation, which occurs independently of the BRAF and NRAS mutation status [26], ensures a sufficient supply of phospholipids essential for proliferation [25]. Besides FA synthesis, FA oxidation (β-oxidation) is associated with melanoma progression. In melanoma patients, increased expression of β-oxidation enzymes has been correlated with poor overall survival [27]. The enzyme carnitine palmitoyltransferase 2 (CPT2), necessary for the translocation of long-chain FAs into the mitochondrial matrix for β-oxidation, has been identified as one of the most significantly upregulated genes in melanoma [28]. Furthermore, elevated mitochondrial function of metastatic melanoma cells was associated with the oxidation of both palmitoylcarnitine and succinate, a TCA cycle intermediate, relative to non-metastatic control cells [29]. Levels of palmitoylcarnitine as well as carnitine and propionylcarnitine were significantly increased in metastatic compared to primary melanoma cells [30].

In sum, melanoma viewed as a metabolic disease can feature highly diverse metabolic signatures, ranging from glycolytic or glutaminolytic to lipogenic/lipolytic phenotypes.

Investigating cancer metabolism using metabolomics contributes to a better understanding of the metabolic vulnerabilities in cancer and helps to facilitate personalized therapies [31,32]. Two approaches, untargeted and targeted metabolomics, are used for metabolic phenotyping. Untargeted metabolomics deals with the comprehensive analysis of all detectable metabolites in a sample, including both known and unknown ones, whereas targeted metabolomics measures a defined group of chemically characterized and biologically annotated metabolites [33,34]. Despite the broad coverage of the metabolome, untargeted analysis lacks the use of reference standards and thus does not allow the absolute quantification of metabolites, thereby hindering the measurement of normal metabolite levels. However, untargeted approaches offer the opportunity to discover novel biomarkers in study samples and can be used to generate new hypotheses [35]. Targeted metabolomics uses reference standards for metabolite identification, making it more sensitive and accurate compared to untargeted metabolomics, thereby allowing absolute quantification of metabolites of interest [34]. Numerous studies of various cancer types have used both untargeted and targeted metabolomics for the discovery of biomarkers or the evaluation of cancer therapies [36,37,38,39].

The aim of the present work was to characterize the metabolomes of different genetic subtypes of melanomas using high-throughput metabolomics in order to identify potential new biomarkers and targets for melanoma therapy. Therefore, we generated tumor xenografts in mice using BRAF mutant A375 and WM47, NRAS mutant WM3000, and triple-wildtype WM3311 human melanoma cells. We analyzed the metabolic signature of xenograft tumors and of plasma obtained from melanoma-bearing mice and healthy mice as a control. Tumor metabolic signatures differed greatly between the four melanoma subtypes irrespective of the genetic mutation. In plasma, several metabolites seemed to be uniquely regulated by the respective tumors, whereas consistently upregulated metabolites in melanoma-bearing versus healthy mice were proposed as potential melanoma biomarkers.

## 2. Results

### 2.1. Metabolite Profiling in Xenograft Tissue

Using high-throughput targeted metabolomics, we compared A375, WM47, WM3000, and WM3311 melanoma xenografts against each other. The analysis revealed 61 hydrophilic/polar metabolites quantified in at least one of the four experimental groups, including 20 amino acids, 22 amino acid-related metabolites, 3 bile acids, 7 biogenic amines, 4 carboxylic acids, 1 cresol, 1 vitamin/cofactor, 1 indole derivative, and 2 nucleobase-related metabolites. Univariate one-way ANOVA revealed 54 significant hits, with at least two of the four groups being significantly different (false discovery rate (FDR)-corrected *p*-value <0.05) (Appendix A). Hierarchical clustering analysis represented in a heatmap illustrates clear sample clustering within each of the four melanomas (Figure 1A), suggesting distinct metabolic profiles of hydrophilic/polar metabolites among the four groups. Principal component analysis (PCA) revealed separation between the A375 and WM47 melanomas, while the WM3000 and WM3311 melanomas overlapped with each other and partly with the A375 and WM47 melanomas. Partial least squares-discriminant analysis (PLS-DA) intensified the clustering (Figure 1B). The results of the cross-validation (Q2 = 0.83; R2 = 0.90; accuracy = 1.0) and the permutation testing (*p* < 0.0005 for 2000 permutations) revealed that the PLS-DA model was of good and predictive quality and that the observed separation was not due to chance. The PLS-DA variable importance in projection (VIP) score, which ranks metabolites according to their importance for group separation, revealed 22 metabolites with a VIP-score >1, including ten amino acid-related metabolites (5-aminovaleric acid, creatinine, phenylacetylglycine, betaine, anserine, sarcosine, ornithine, methionine sulfoxide, alpha-aminobutyric acid, carnosine), four amino acids (aspartate, glutamine, alanine, threonine), three biogenic amines (beta-alanine, histamine, gamma-aminobutyric acid), two bile acids (taurocholic acid and tauromuricholic acid), 1 carboxylic acid (succinate), one indole derivative (indoxyl sulfate), and 1 cresol (*p*-cresol-sulfate) (Figure 1C).

Discriminating metabolites with a VIP score >1 and an FDR-corrected *p*-value < 0.05 were further used for metabolic pathway analysis (MetPA) to identify major metabolic pathways that were altered between the four melanoma subtypes. Thus, 21 metabolites (VIP score >1, FDR-corrected *p*-value < 0.05; all mentioned above except *p*-cresol sulfate; Appendix A) were uploaded to the web-based MetaboAnalyst 4.0 tool for pathway analysis. MetPA revealed three metabolic pathways, namely beta-alanine metabolism (impact score = 0.46), alanine, aspartate, and glutamate metabolism (impact score = 0.42), and histidine metabolism (impact score = 0.33), that were altered between melanoma subtypes with statistical significance (*p* < 0.05) based on pathway enrichment analysis and high impact scores based on pathway topological importance (Figure 2) (Appendix A).

Analysis of the lipids and lipid-like metabolites included in our targeted metabolomics approach revealed 215 metabolites quantified in at least one of the four experimental groups, including 15 ceramides (Cer), 1 dihydroceramide (DH-Cer), 12 hexosylceramides (HexCer), 7 dihexosylceramides (Hex2Cer), 4 trihexosylceramides (Hex3Cer), 63 phosphatidylcholines (PC), 10 lyso-phosphatidylcholines (lyso-PC), 7 FAs, 32 acylcarnitines (AC), 4 diglycerides (DG), 46 triglycerides (TG), and 14 sphingomyelins (SM). One-way ANOVA resulted in 211 significant hits, with at least two of the four groups being significantly different (FDR-corrected *p*-value < 0.05) (Appendix A). The results of the hierarchical clustering analysis (heatmap) illustrate a perfect group separation between the four melanomas (Figure 3A). This separation suggests that the lipid profiles were highly distinct among the four groups. PCA as well as PLS-DA (Q2 = 0.96; R2 = 0.98; accuracy = 1.0; *p* < 0.0005 for 2000 permutations) showed excellent group separation between the melanoma xenografts (Figure 3B). The PLS-DA VIP score revealed that discriminating metabolites with a VIP score > 1 consisted of 44% Cer species (18% Cer, 18% HexCer, 7% Hex2Cer, 2% Hex3Cer), 25% TG, 21% PC species (14% PC, 7% lyso-PC), 5% AC, 4% SM, and 2% DG (Figure 3C).

Individual PCA of each lipid class demonstrated that, Cer species (Cer, DH-Cer, HexCer, Hex2Cer, Hex3Cer) and PC species (PC, lyso-PC) indeed define the distinct lipid metabolomes of A375, WM47, WM3000, and WM3311 melanomas (Figure 4A,B). Most of both the Cer and PC species were higher in WM47 and WM3311 melanomas compared to A375 and WM3000 melanomas (Appendix A). Many FAs and ACs were higher in WM47 and WM3000 melanomas compared to A375 and WM3311 melanomas, which contributes to the observed separation of WM47 and WM3000 from A375 and WM3311 tissue samples in the PCA (Figure 4C and Appendix A). Interestingly, we found that the xenografts differed in the percentage of tumor necrosis, with WM47 and WM3000 xenografts presenting low levels of necrosis compared to A375 and WM3311 xenografts (Appendix A). Correlation analysis resulted in a negative correlation between several ACs and tumor necrosis (Appendix A). However, the level of necrosis did not correlate with the size of the melanoma xenografts. The discrimination of WM3311 melanoma xenografts from A375, WM47, and WM3000 melanomas was based on the high abundance of TGs in WM3311 melanoma tissue compared to the other three melanoma tissues, in which most TGs were below the limit of detection (LOD) (Figure 4D and Appendix A). Moreover, SMs were differentially expressed in the four melanoma subtypes as well (Figure 4E and Appendix A).

### 2.2. Metabolite Profiling in Plasma

Next, we analyzed metabolic signatures in plasma from melanoma-bearing mice compared to plasma from non-tumor mice (NTM). The metabolomics analysis resulted in 66 hydrophilic/polar metabolites quantified in at least one of the five experimental groups, including 20 amino acids, 24 amino acid-related metabolites, 6 bile acids, 6 biogenic amines, 4 carboxylic acids, 1 cresol, 1 vitamin/cofactor, 3 indole derivatives, and 1 alkaloid. Additionally, we added blood glucose levels and concentrations of the ketone body beta-hydroxybutyrate to the analysis because we monitored those parameters during the in vivo experiment. One-way ANOVA revealed 52 significant hits, with at least two of the five groups being significantly different (FDR-corrected *p*-value < 0.05) (Appendix A). NTM were clustered separately from the melanoma xenografts (Figure 5A), which was also reflected in the PLS-DA analysis (Q2 = 0.62; R2 = 0.76; accuracy = 0.91; *p* < 0.0005 for 2000 permutations) (Figure 5B). In addition, clustering of plasma samples from mice bearing melanoma xenografts of different genetic backgrounds was observed, indicating distinct metabolic signatures of hydrophilic/polar plasma metabolites in the melanoma subtypes. The PLS-DA VIP score resulted in eight metabolites with a VIP-score > 1, including the three bile acids deoxycholic acid, taurodeoxycholic acid, and cholic acid as well as *p*-cresol sulfate, proline betaine, succinate, 5-aminovaleric acid, and carnosine. Concentrations of taurodeoxycholic acid, cholic acid, *p*-cresol sulfate, proline betaine, and succinate were lowest in NTM and upregulated to different extents in the plasma of the four melanoma-xenograft subtypes (Figure 5C).

To identify which metabolic pathways were altered in plasma by the different melanoma xenografts, we performed MetPA for A375 vs. NTM, WM47 vs. NTM, WM3000 vs. NTM, and WM3311 vs. NTM. For each MetPA of plasma metabolites, we considered metabolites with an FDR-corrected *p*-value < 0.05 and a fold change (FC) > 1.5 (Appendix A). Thus, we obtained a list of metabolites, either up- or downregulated, for each melanoma–NTM comparison and uploaded the respective metabolite concentration table into the pathway analysis tool of MetaboAnalyst 4.0. MetPA revealed beta-alanine metabolism to be significantly upregulated, with a high impact score in all melanomas vs. NTM (Figure 6A–D). Interestingly, the analysis in melanoma xenograft tumors also ranked beta-alanine metabolism as having the highest impact value among the four melanomas (Figure 2), thus highlighting beta-alanine metabolism as a significant contributor to melanoma pathophysiology. Glutamine and glutamate as well as taurine and hypotaurine metabolism were identified by MetPA to be differentially regulated in A375 and WM3311 versus NTM (Figure 6A,D). In plasma of WM3311 versus NTM, alanine, aspartate and glutamate metabolism was additionally ranked with a significant p-value and the highest impact score (Figure 6D) (Appendix A).

The analysis of lipids and lipid-like metabolites in plasma resulted in 390 metabolites quantified in at least one of the five experimental groups, including 12 Cer, 1 DH-Cer, 9 HexCer, 6 Hex2Cer, 6 Hex3Cer, 68 PC, 12 lyso-PC, 8 FA, 32 AC, 3 DG, 209 TG, 14 SM, and 10 cholesteryl esters (CE). One-way ANOVA resulted in 317 significant hits, with at least two of the five groups being significantly different (FDR-corrected *p*-value <0.05) (Appendix A). Clustering analysis revealed partial rearrangements of plasma samples between groups based on lipids and lipid-like metabolites (Figure 7A, Appendix A). Separation of A375/WM3000 melanomas from NTM and WM3311/WM47 melanomas was indicated by PLS-DA analysis (Q2 = 0.61; R2 = 0.87; accuracy = 0.95; *p* < 0.0005 for 2000 permutations) (Figure 7B). This was also reflected in the heatmap of the VIP score analysis, which showed that several metabolites were downregulated in plasma of A375 and, especially, WM3000 melanoma-bearing mice compared to the remaining groups (Figure 7C). Discriminating metabolites with a VIP score >1 consisted of 70% TG, 10% AC, 9% PC, 4% HexCer, 3% Cer, 2% CE, 2% FA, and 1% lyso-PC (Figure 7C). PCA of lipid subclasses in plasma indicated that PC species (PC, lyso-PC) were most likely responsible for the separation of plasma derived from melanoma-bearing mice versus NTM (Appendix A).

We further analyzed which metabolites were uniquely up- and downregulated in plasma of one of the melanoma xenograft types compared to NTM and which metabolites overlapped between two, three, or all four melanomas versus NTM. The Venn diagram summarizes intersections among the four set lists (A375 vs. NTM, WM47 vs. NTM, WM3000 vs. NTM, and WM3311 vs. NTM) including only plasma metabolites with an FDR-corrected *p*-value < 0.05 and FC >1.5 (Figure 8, Appendix A). Of those metabolites, 68, 7, 7, and 23 were solely upregulated in plasma of A375, WM47, WM3000, and WM3311 melanoma-bearing mice compared to NTM, respectively (Figure 8A). These results indicate a strong and unique contribution of the respective melanoma subtype on the host plasma metabolome. Unique downregulations of metabolites were mostly identified in plasma of WM47 (22 metabolites) and WM3000 melanoma-bearing mice (31 metabolites) versus NTM (Figure 8B). Considering that A375 and WM47 cells carry the same BRAF mutation, it is striking that the metabolite regulation in plasma differed so substantially between these two xenograft models (A375: 68 upregulated and 4 downregulated vs. WM47: 7 upregulated and 22 downregulated metabolites). On the other hand, the 17 upregulated and the 2 downregulated plasma metabolites commonly detected in both BRAF-mutated melanoma xenograft variants versus NTM could be due to the BRAF mutation and associated oncogenic signaling. However, this approach of filtering metabolites revealed that seven metabolites were consistently upregulated in melanomas versus NTM (Figure 8A), namely beta-alanine, *p*-cresol sulfate, sarcosine, tiglylcarnitine, Hex2Cer (d18:1/16:0), Hex2Cer(d18:1/20:0), and PC ae C42:4. Importantly, those seven metabolites were elevated in plasma of melanoma-bearing mice regardless of tumor size. In contrast, no matching downregulated metabolite was identified (Figure 8B).

### 2.3. Biomarker Analysis

In the first part of the analysis, we analyzed metabolic processes involved in melanoma pathophysiology to better understand the metabolic heterogeneity of melanoma subtypes. Additionally, we attempted an exploratory plasma biomarker analysis to identify potential biomarkers that distinguish melanomas, regardless of their genetic background and metabolic flexibility, from healthy controls. Thus, we analyzed the predictive performance of the seven identified metabolites which were consistently upregulated in plasma from melanoma-bearing mice compared to NTM (Figure 8). Therefore, we pooled the data of all plasma samples obtained from melanoma-bearing mice (*n* = 46) versus NTM (*n* = 9) for receiver operating characteristic (ROC) curve analysis. Individual ROC curve analysis suggested that the seven metabolites might potentially serve as biomarkers for melanoma based on the obtained area under the ROC curve (AUROC), which indicates how well the chosen candidate biomarker can distinguish between two diagnostic groups (healthy/disease), and the sensitivity as well as specificity values (Table 1, Appendix A).

Moreover, we combined the seven biomarker candidates to develop biomarker models based on well-established machine learning and statistical algorithms (linear support vector machine (SVM), PLS-DA, random forests and logistic regression). The performance of each seven-biomarker model created using a linear SVM with an AUROC of 0.967 (95% CI: 0.8–1), PLS-DA with an AUROC of 0.92 (95% CI: 0.746–1), random forests with an AUROC of 0.991 (95% CI: 0.954–1), or logistic regression with an AUROC of 0.832 (95% CI: 0.457–0.996) indicated a strong discriminative ability for the seven selected metabolites in screening for melanoma (Figure 9A–D left panel). For each model, the average of the predicted class probabilities of each sample across 100 cross-validations is shown in the respective prediction overview and summarized in the corresponding confusion matrix (Figure 9A–D right panel). Best discrimination was achieved by the model algorithm based on random forests that classified 45 of 46 plasma samples from melanoma-bearing mice and all nine NTM plasma samples correctly, suggesting that the metabolic signature consisting of those seven metabolites could indeed distinguish between melanoma and healthy at least in preclinical melanoma models.

## 3. Discussion

Reprogrammed metabolism is a hallmark of cancer and has been linked to cancer metastasis, drug resistance, and patient survival [40]. To understand how metabolism is rewired in melanoma, previous metabolomics studies using various analytical techniques focused on the identification of metabolic profiles in melanoma tissue samples, biofluids or cell lines. Increased amino acids, especially glutamine and glutamate, and decreased glucose levels have been associated with melanoma metastasis [29,30,41,42,43,44,45,46]. Metabolites related to protein methylation correlated with metastatic capacity of human melanoma xenografts [47]. Several studies consistently reported that lipid profiles differed between metastatic and non-metastatic melanoma [30,41,42,43,44,48,49,50,51,52]. For instance, Kim et al. associated increased levels of phosphatidylinositol species with the metastatic potential of melanoma cells, proposing these lipids as possible biomarkers [43]. Moreover, it was shown recently that melanoma cells undergo metabolic disruptions when exposed acutely to mutation-specific inhibitors. However, long-term exposure reversed these metabolic shifts to basal metabolism, indicating that melanoma cells acquired resistance capabilities [53].

We propose seven plasma metabolites as potential melanoma biomarkers, including beta-alanine, *p*-cresol sulfate, sarcosine, tiglylcarnitine, Hex2Cer (d18:1/16:0), Hex2Cer (d18:1/20:0), and PC ae C42:4, to distinguish healthy from melanoma-bearing mice. Of those seven metabolites, sarcosine is a well-established urine biomarker for prostate cancer [54,55,56]. However, to our knowledge, sarcosine has not yet been reported as biomarker for melanoma.

In agreement with our findings, Ang et al. reported upregulation of PCs, including PC ae C42:4, in plasma of mice bearing BRAF mutant melanoma xenografts, which was reversed by MEK inhibitor treatment [57]. Interestingly, some of the identified PCs served as predictive markers for the MEK inhibitor response in a phase I clinical trial [57]. Muqaku et al. compared serum metabolite profiles between melanoma patients with low and high tumor loads and control individuals. In patients with high tumor load, long-chain, highly unsaturated PCs, including PC ae C42:4, were significantly decreased, whereas concentrations of 39 ACs, including tiglylcarnitine, were significantly increased, which indicated extensive lipolysis associated with cachexia in these patients [49]. Another study that performed metabolite profiling of serum samples suggested five metabolites, including one PC (PC aa C40:3) and three ACs (carnitine, octanoylcarnitine and methylmalonylcarnitine), as well as ethanol, as diagnostic biomarkers for advanced-stage melanoma [58]. In those three studies, metabolomics analysis was performed using the AbsoluteIDQ^®^ p180 kit (Biocrates, Innsbruck, Austria) which enables the quantification of about 180 metabolites. In addition, Bayci et al. performed a nuclear magnetic resonance (NMR)-based analysis to expand the metabolomics approach [58]. To capture an even broader spectrum of metabolites compared to previous studies, we used the MxP^®^ Quant 500 kit combined with the AC assay from Biocrates capable of quantifying more than 600 metabolites from 26 compound classes for metabolite profiling of melanoma-bearing mice. A great advantage of the AC assay is its increased acylcarnitine panel beyond that of the AbsoluteIDQ^®^ p180 or the MxP^®^ Quant 500 kit. Using the AC assay, isomeric and isobaric groups of ACs can be separated, leading to a more accurate AC profiling.

In the present analysis, beta-alanine metabolism was significantly increased in plasma of each melanoma subtype compared to healthy controls. Emerging evidence suggests beta-alanine as a prognostic biomarker in breast cancer tissue [59,60]. Moreover, elevated levels of beta-alanine in various biofluids were reported for prostate, colon, and head and neck cancer patients [61,62,63,64]. Beta-alanine is not included in the AbsoluteIDQ^®^ p180 kit used by Ang et al., Muqaku et al. and Bayci et al., which may be why we were able to identify beta-alanine as a potential plasma biomarker for melanoma for the first time.

Metabolite profiling of different melanoma xenografts revealed highly distinct lipid profiles. Emerging evidence supports the importance of lipid metabolism in melanoma progression since many alterations of the lipid metabolic network contributing to sustained cell proliferation and melanoma metastasis have been reported [21]. Lipids constitute an important energy source for cells but are also structural components of cellular membranes. Major cell membrane lipids, such as Cer and PCs, are bioactive compounds involved in a variety of cellular processes [65,66]. Complete separation of the melanoma subtypes was observed regarding Cer species (Cer, DH-Cer, HexCer, Hex2Cer, Hex3Cer). Cer have been reported to be involved in differentiation, proliferation, growth arrest, and apoptosis [67]. Several studies have showed that melanoma cells are able to rearrange sphingolipid metabolism by preventing intracellular Cer accumulation by induction of Cer-degrading enzymes and downregulation of Cer-promoting enzymes [68]. Thus, increasing the intracellular levels of Cer was shown to reduce cancer cell proliferation [69,70,71,72,73,74,75]. Besides Cer, PC species (PC, lyso-PC) gave excellent separation of melanoma subtypes in the present study. Aberrant glycerophospholipid metabolism has been related to melanoma metastatic potential [43,48]. Moreover, combined desorption electrospray ionization mass spectrometry imaging and lipidome analyses revealed heterogeneous phospholipid composition in zebrafish melanomas, including dysregulated glycerophospholipid metabolism with increased PC and phosphatidylethanolamine (PE) levels [76].

Increased lipolysis and β-oxidation have been associated with melanoma progression [21,27,28]. Most ACs were found to be higher in WM47 and WM3000 compared to A375 and WM3311 xenografts, indicating increased β-oxidation in WM47 and WM3000 melanomas. Interestingly, several ACs negatively correlated with the percentage of necrosis seen in histological staining of the melanoma xenografts, suggesting that if enough ACs are available to sufficiently drive β-oxidation, then the energy requirements of cells are fulfilled and allow them to proliferate. On the other hand, if the high energy demand of cancer cells is fettered by insufficient energy production due to low availability of ACs, cells might undergo metabolic stress associated with necrosis [77].

Regarding hydrophilic/polar metabolites in melanoma xenografts, metabolic profiling revealed strong separation of A375 and WM47 xenograft samples, whereas the WM3000 and WM3311 xenograft samples tended to be clustered but separated from the distinct A375 and WM47 melanomas. The observed profile separation between the different BRAF mutant melanoma xenografts supports the hypothesis of metabolic heterogeneity irrespective of the genetic driver mutation. Moreover, several plasma metabolites were uniquely regulated in either A375 or WM47 melanoma samples, whereas only a few metabolites were consistently up- or downregulated in plasma samples from both BRAF mutant melanomas compared to NTM. A possible reason for the distinct metabolite profiles between the BRAF mutant melanoma xenografts could be that A375 and WM47 cells originate from primary and metastatic melanomas, respectively. Moreover, genetic alterations beyond the BRAF mutation could contribute to individual metabolic regulatory differences.

Metabolic heterogeneity between melanoma subtypes was also reflected by significant alterations in beta-alanine, alanine, aspartate, glutamate, as well as histidine metabolism. In a previous study, altered glutamate and alanine levels were identified as the main differences in conditioned media of cell cultures of early-stage non-metastatic and metastatic melanoma cells, indicating a stage-specific requirement for glutamate and alanine for growth and aggressiveness [78]. Changes in alanine, aspartate, glutamate, as well as beta-alanine metabolism were recently associated with melanoma development [43]. Interestingly, some studies proposed anti-tumor effects, including reduced proliferation, migration, and metabolic rate, of beta-alanine on different cancer types [79,80,81]. However, the biological effects of beta-alanine on melanoma remain a subject for further investigation.

Taken together, melanoma xenografts presented highly heterogeneous metabolic signatures in xenograft tissue. Nevertheless, we can propose seven consistently upregulated metabolites in melanoma versus healthy mice as potential melanoma biomarkers.

## 4. Materials and Methods

### 4.1. Cell Lines

The melanoma cell lines A375 (BRAFV600E/NRASwt; Sigma Aldrich, Darmstadt, Germany), WM47 (BRAFV600E/NRASwt; Wistar Institute, Philadelphia, PA, USA), WM3000 (BRAFwt/NRASQ61R; Rockland Inc., Philadelphia, PA, USA), and WM3311 (BRAF/NRAS/NF1 wild-type; Wistar Institute, Philadelphia, PA, USA) were cultured as described previously [10] and used to establish human melanoma xenografts.

### 4.2. Animals and Sample Collection

The in vivo experiments were performed in accordance with the Salzburg Animal Care and Use Committee (Study approval no. 20901-TVG/112/6-2018). Animals were maintained under specific pathogen-free conditions and care conformed to the Austria Act on Animal Experimentation. Mice had ad libitum access to water and chow. A total of 1 × 107 cells in 200 µL of a 1:1 mixture of matrigel (Corning, New York, NY, USA) and serum-free medium were subcutaneously injected into the right flank of 5- to 7-week-old female CD-1 nude mice (*n* = 10–13) (Charles River, Sulzfeld, Germany). Control animals (*n* = 9) were kept tumor-free. Blood glucose and beta-hydroxybutyrate levels were monitored over time (Precision Xceed System, Abbott, Chicago, IL, USA). For metabolomics analysis, blood was taken by cardiac puncture and collected in MiniCollect^®^ Lithium Heparin tubes (Greiner Bio-One, Kremsmünster, Austria) and centrifuged for 3 min at 2000 g. Plasma samples were stored at −80 °C until metabolomics analysis. Tumor size at the day of collection is shown in Appendix A. When tumors were harvested, one half of the tumor was snap frozen in liquid nitrogen for metabolomics analysis while the other half was formalin-fixed and paraffin-embedded (FFPE) for histological analysis.

### 4.3. Histology

Hematoxylin and eosin (HE) staining was performed using 4-μm FFPE sections of the xenografts. After deparaffination, rehydrated tissue sections were incubated for 6 min in Mayer′s hemalum solution (Merck KGaA, Darmstadt, Germany). Counterstaining was performed with two dips in 0.25% Eosin Y (Merck KGaA, Darmstadt, Germany) solution in 70% ethanol. Slides were mounted with Histokitt (Karl Hecht GmbH & Co KG, Sondheim vor der Rhön, Germany). The percentage of the necrotic area in the tumor sections was scored from 0–100% by three examiners and averaged.

### 4.4. Targeted Metabolomics

To capture a broad spectrum of metabolites, we used the MxP^®^ Quant 500 kit (Biocrates Life Sciences AG, Innsbruck, Austria), capable of quantifying more than 600 metabolites from 26 compound classes, following the manufacturer’s instructions. All reagents, internal standards (IS), calibration standards, quality controls (QCs), and the test mix required for the MxP^®^ Quant 500 analysis were included in the kit. In addition, ACs were analyzed using the inhouse UHPLC-MS/MS-based AC assay from Biocrates, capable of separating different isomeric and isobaric groups of ACs. Isotope-labeled IS were used and the closest eluting deuterated IS were assigned for those ACs lacking their own isotope-labeled IS.

Samples were prepared the same way for both assays. Briefly, 50–80 mg of tumor tissues were transferred to 0.5 mL Precellys^®^ vials followed by the addition of 85:15 ethanol:0.01 M phosphate as a lysis buffer with a volume (µL) of three times the tissue weight (mg). Tissue homogenization was carried out at 4 °C and 5800 rpm using a Precellys^®^ 24 homogenizer coupled to a Cryolys^®^ cooling unit (Bertin Instruments, Bretonneux, France). The homogenates were centrifuged at 10,000 g for 2 min at 2–4 °C. Supernatants were collected and stored at −80 °C until analysis. Plasma samples as well as tissue homogenate supernatant were thawed on ice, then vortexed before use.

#### 4.4.1. MxP^®^ Quant 500 Kit

10 µL of calibration standards, QCs, tissue homogenates, or plasma samples were added onto the respective filter spots of the 96-well-based Biocrates sample preparation plate which was impregnated with internal standards. After drying the samples under nitrogen, 50 µL of 5% phenyl-isothiocyanate solution was added in each well for derivatization. After incubation for 1 h and subsequent drying under nitrogen, 300 µL 5 mM ammonium acetate in methanol was added for metabolite extraction. The extract was diluted with defined volumes of running solvent according to the manufacturer′s protocol. The extracts were analyzed using a Waters ACQUITY UPLC coupled with a Waters TQ-S MS, using electrospray ionization and multiple reaction mode. Amino acids, biogenic amines, and other small molecules were analyzed by liquid chromatography (LC) in two different injections and lipids were analyzed by flow injection analysis (FIA. Depending on the sample matrix, the assay could quantify up to 20 amino acids, 30 amino acid-related metabolites, 14 bile acids, 9 biogenic amines, 7 carboxylic acids, 12 fatty acids, 4 hormones, 4 indoles, 2 nucleobases, and 1 cresol, alkaloid, amine oxide, and vitamin/cofactor by LC, whereas metabolites quantified by FIA include 28 Cer, 8 DH-Cer, 19 HexCer, 9 Hex2Cer, 6 Hex3Cer, 74 PC, 14 lyso-PC, 14 SM, 22 CE, 44 DG, 242 TG, and 1 hexose.

#### 4.4.2. AC Assay

20 µL of calibration standards, QCs, tissue homogenates, or plasma samples were added onto the respective filter spots of the 96-well-based Biocrates sample preparation plate which was impregnated with internal standards, followed by drying the samples under gentle nitrogen flow at 4 bar. For AC extraction, 100 µL methanol was added and samples were incubated with shaking at 600 rpm for 10 min at room temperature followed by centrifugation for 2 min at 500 g. The addition of 100 µL methanol, incubation, and centrifugation was repeated to ensure the complete extraction of ACs. After extraction, the extracts were dried at 50 °C for 25 min under nitrogen flow at 4 bar. Prior to injection, the dried extracts were reconstituted with 50 µL of ice-cold 50% methanol (v/v) in Milli-Q water. Then, 2 µL of reconstituted extract solution was injected. The extracts were analyzed using a Dionex UltiMate 3000 UPLC system coupled to a TSQTM Vantage MS. All the measurements were carried out in positive ion multiple reaction mode. Use of IS and the quantitative analysis by comparing the analyte and its corresponding IS corrected the matrix effect and compensated for any possible variation during the entire process of sample preparation and analysis. The accuracy and precision of the method in each set of analyses were evaluated using calibrators. Depending on the sample matrix, the assay could quantify up to 44 ACs, including short-chain, medium-chain, long-chain, branched-chain, isomeric, and isobaric compounds.

### 4.5. Data Processing and Statistical Analysis

Raw data of the MxP^®^ Quant 500 assay were exported and quantified using the Biocrates MetIDQ^TM^ software. Raw data files of the AC assay were generated by Xcalibur software (Thermo Electron Corp. Waltham, MA, USA) and exported to MetIDQ^TM^ for further analysis, including peak integration, retention time correction, area calculation, calibration curve preparation, and concentration calculation. Data for tumor tissue were normalized using the tissue factor in MetIDQ^TM^. Quality control samples-based data normalization was performed to minimize the variation of analyses. Initial data cleaning was performed by excluding metabolites with >20% missing values or values below the LOD in all experimental groups. Thus, all metabolites with >80% of the concentration values above the LOD in at least one of the four experimental groups were included for statistical analysis. Remaining missing values were replaced by 1/5 of the minimum positive value of each variable. Data were log-transformed to assume normal distribution before comprehensive downstream analysis using the web-based tool MetaboAnalyst 4.0 (https://www.metaboanalyst.ca) [82]. For both tissue and plasma data, significant changes in metabolite levels were identified by one-way ANOVA followed by Fisher′s LSD post hoc test. To correct for multiple comparisons and thus to minimize false positives, FDRs were calculated based on the Benjamini–Hochberg procedure. FDR-corrected *p*-values < 0.05 were considered statistically significant. For plasma data, FCs were calculated to evaluate differences between metabolites in plasma obtained from melanoma-bearing mice compared to non-tumor mice. FCs > 1.5 were considered important. Both unsupervised PCA and supervised PLD-DA were performed whenever necessary to determine the metabolic signature contributing to group separation. PLS-DA decreases intergroup variability and improves separation. However, PLS-DA is prone to data overfitting. Thus, the quality of the model was assessed by cross-validation (calculation of Q2, R2, and accuracy values) and the overfitting tendency of the model was validated using permutation testing. The PLS-DA VIP-score was calculated and metabolites with a VIP score > 1 were considered important for group separation. To increase the robustness of our analyses, we combined the ANOVA and PLS-DA VIP score as well as the ANOVA and FC results in order to determine a reliable list of metabolites that significantly contributed to the separation of the four melanoma xenografts and each melanoma from the non-tumor controls, respectively. Thus, for MetPA we only considered metabolites that overlapped between the two different statistical approaches (FDR-corrected *p*-value < 0.05 and VIP score > 1 or FDR-corrected *p*-value < 0.05 and FC > 1.5). The Homo sapiens and Mus musculus KEGG pathway libraries were used as references for the MetPA of human melanoma cell xenograft tissue metabolites and mouse plasma metabolites, respectively. For exploratory biomarker analysis, we filtered metabolites (FDR-corrected *p*-value < 0.05 and FC > 1.5) that were consistently up- or downregulated in plasma samples from melanoma-bearing mice compared to healthy controls. ROC curve analysis was performed on biomarker candidates individually or combined to evaluate the predictive performance of the potential biomarkers. Heatmaps were created using MetaboAnalyst 4.0 and Venn diagrams were generated on the http://bioinformatics.psb.ugent.be/webtools/Venn/ website.

## 5. Conclusions

Metabolomics enables investigation of individual or disease-specific metabolic fingerprints and helps in understanding complex metabolic alterations in various diseases. Metabolite profiling of genetically different melanoma subtypes revealed highly heterogeneous lipid profiles in xenograft tissues independent of the main genetic driver. Moreover, beta-alanine metabolism was suggested to play an important role in melanoma metabolism. We proposed seven metabolites in plasma, including beta-alanine, as potential melanoma biomarkers. Clinical evaluation of the identified potential biomarkers will be part of our future analyses. Moreover, our results suggest lipid metabolism as well as beta-alanine metabolism as highly interesting targets for melanoma therapy.

## Figures and Tables

**Figure 1 cancers-13-00434-f001:**
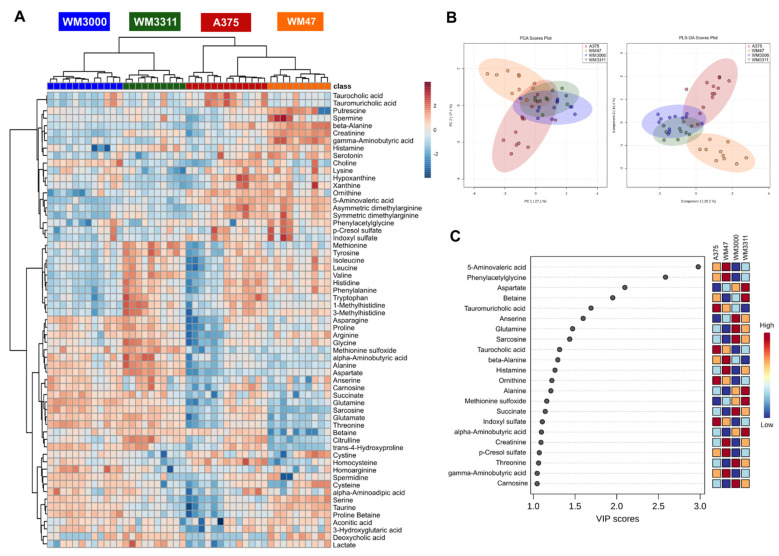
Metabolic analysis of hydrophilic/polar metabolites in melanoma xenografts. (**A**) Heatmap, (**B**) principal component analysis (PCA) and partial least squares-discriminant analysis (PLS-DA) scores, and (**C**) PLS-DA variable importance in projection (VIP) score derived from the hydrophilic/polar metabolite profiling data of A375, WM47, WM3000, and WM3311 melanoma xenografts; *n* = 10–13.

**Figure 2 cancers-13-00434-f002:**
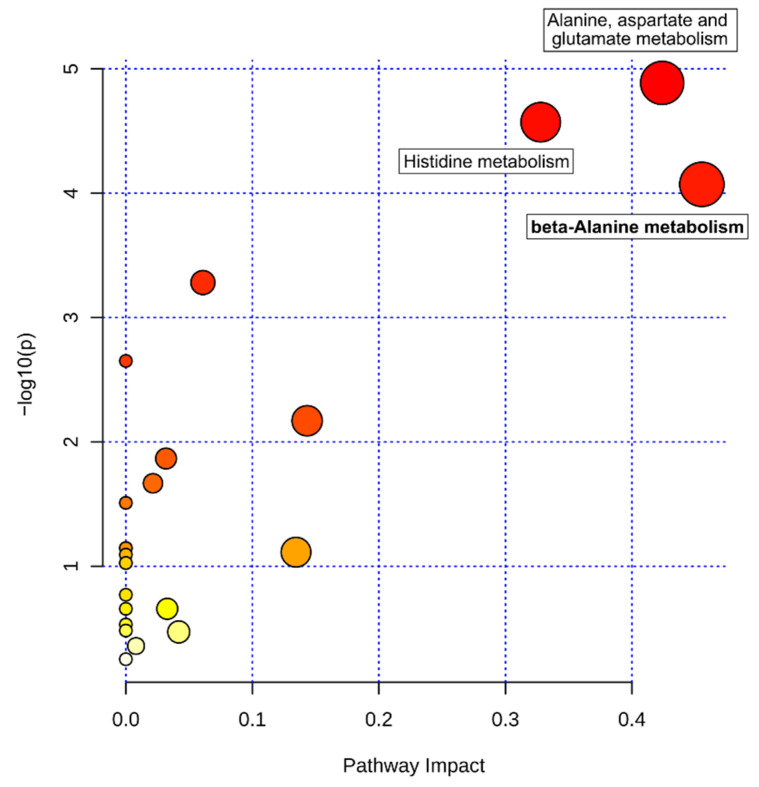
Metabolic pathway analysis of tumor metabolites. The metabolome view shows matched pathways arranged by *p*-values from pathway enrichment analysis (Y-axis) and pathway impact values from pathway topology analysis (X-axis). Node color and radius are based on the *p*-value and pathway impact value, respectively.

**Figure 3 cancers-13-00434-f003:**
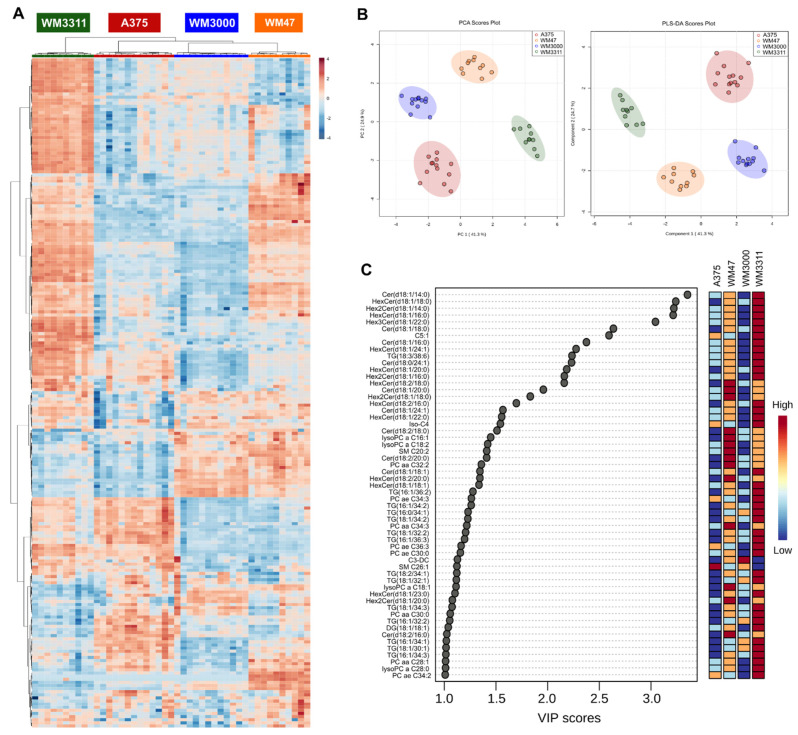
Metabolic analysis of lipids and lipid-like metabolites in melanoma xenografts. (**A**) Heatmap, (**B**) PCA and PLS-DA scores, and (**C**) PLS-DA VIP score derived from the lipid and lipid-like metabolite profiling data of A375, WM47, WM3000, and WM3311 melanoma xenografts. *n* = 10–13; aa: acyl-acyl, ae: acyl-alkyl, Cx:y:x = number of carbons in the fatty acid side chain and y = number of double bonds in the fatty acid side chain, C3-DC: malonylcarnitine C5:1: tiglylcarnitine, Cer: ceramide, DG: diglyceride, HexCer: hexosylceramide, Hex2Cer: dihexosylceramide, Hex3Cer: trihexosylceramide, Iso-C4: iso-butyrylcarnitine, lysoPC: lyso-phosphatidylcholine, PC: phosphatidylcholine, SM: sphingomyelin, TG: triglyceride.

**Figure 4 cancers-13-00434-f004:**
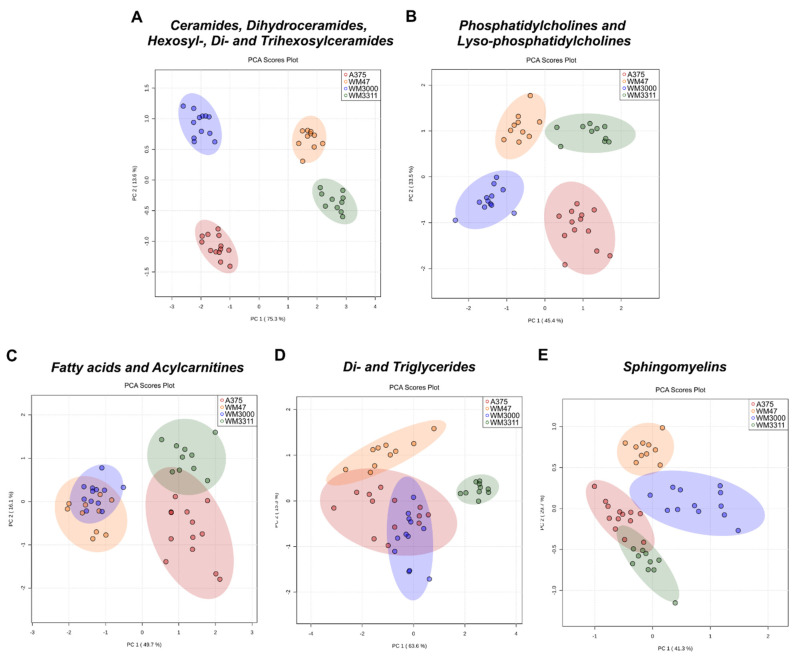
Metabolic analysis of lipid-subclasses in melanoma xenografts. PCA scores derived from lipid and lipid-like metabolite profiling data of A375, WM47, WM3000, and WM3311 melanoma xenografts divided into lipid-subclasses: (**A**) ceramides, dihydroceramides, hexosylceramides, di- and trihexosylceramides, (**B**) phosphatidylcholines and lyso-phosphatidylcholines, (**C**) fatty acids and acylcarnitines, (**D**) di- and triglycerides, and (**E**) sphingomyelins; *n* = 10–13.

**Figure 5 cancers-13-00434-f005:**
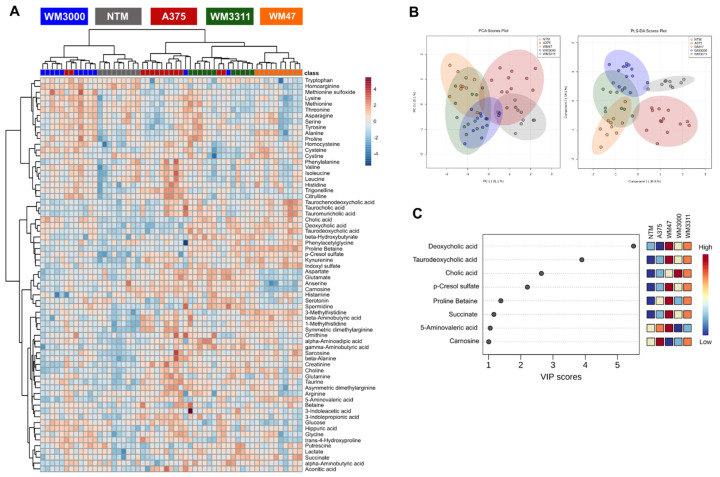
Metabolic analysis of hydrophilic/polar metabolites in plasma. (**A**) Heatmap, (**B**) PCA and PLS-DA scores, and (**C**) PLS-DA VIP score derived from the hydrophilic/polar metabolite profiling data of plasma samples from mice bearing A375, WM47, WM3000, and WM3311 melanoma xenografts and non-tumor mice (NTM); *n* = 9–13.

**Figure 6 cancers-13-00434-f006:**
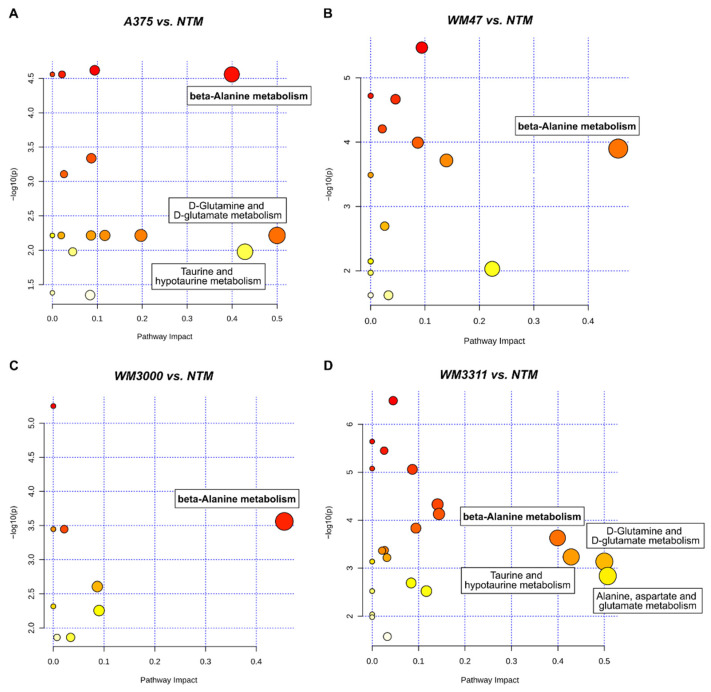
Metabolic pathway analysis of plasma metabolites. For each comparison—(**A**) A375 vs. NTM, (**B**) WM47 vs. NTM, (**C**) WM3000 vs. NTM, and (**D**) WM3311 vs. NTM—the metabolome view shows matched pathways arranged by *p*-values from pathway enrichment analysis (Y-axis) and pathway impact values from pathway topology analysis (X-axis). Node color and radius are based on the *p*-value and pathway impact value, respectively.

**Figure 7 cancers-13-00434-f007:**
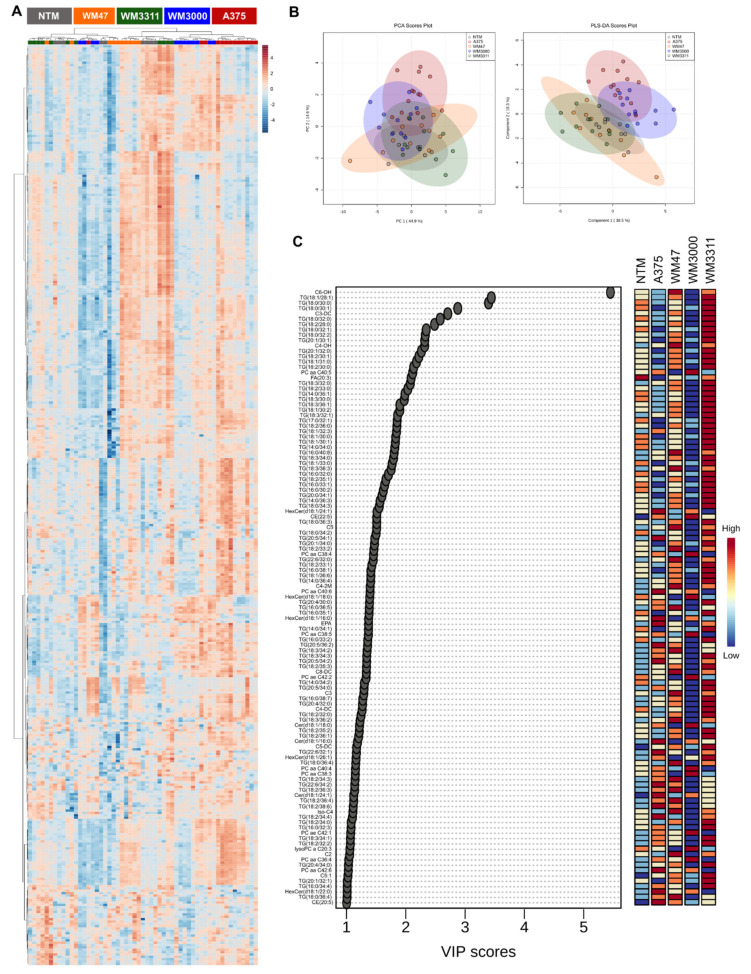
Metabolic analysis of lipids and lipid-like metabolites in plasma. (**A**) Heatmap, (**B**) PCA and PLS-DA scores, and (**C**) PLS-DA VIP score derived from the lipids and lipid-like metabolite profiling data of plasma samples from mice bearing A375, WM47, WM3000, and WM3311 melanoma xenografts and non-tumor mice. *n* = 9–13; aa: acyl-acyl, ae: acyl-alkyl, Cx:y:x = number of carbons in the fatty acid side chain and y = number of double bonds in the fatty acid side chain, C2: acetylcarnitine, C3: propionylcarnitine, C3-DC: malonylcarnitine, C4-2M: 2-methylbutyrylcarnitine, C4-DC: succinylcarnitine, C4-OH: hydroxybutyrylcarnitine, C5: valerylcarnitine, C5-DC: glutarylcarnitine, C5:1: tiglylcarnitine, C6-OH: hydroxyhexanoylcarnitine, C8-DC: suberylcarnitine, CE: cholesteryl ester, Cer: ceramide, EPA: eicosapentaenoic acid, FA (20:3): eicosatrienoic acid, HexCer: hexosylceramide, Iso-C4: iso-butyrylcarnitine, lysoPC: lyso-phosphatidylcholine, PC: phosphatidylcholine, SM: sphingomyelin, TG: triglyceride.

**Figure 8 cancers-13-00434-f008:**
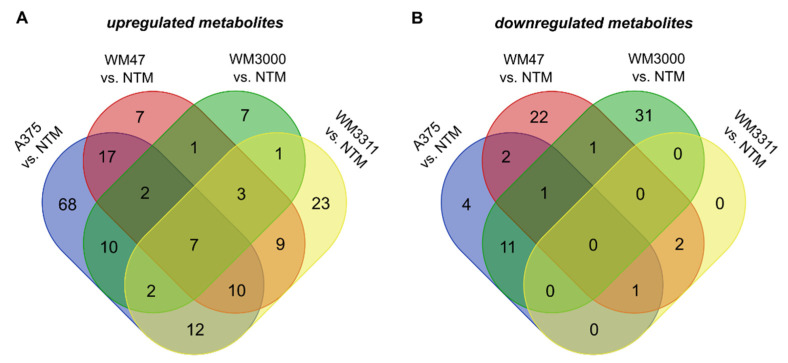
Overlap of plasma metabolites between melanoma-bearing mice and non-tumor mice. The Venn diagram demonstrates the intersections of (**A**) upregulated and (**B**) downregulated plasma metabolites between each type of melanoma xenograft model versus NTM.

**Figure 9 cancers-13-00434-f009:**
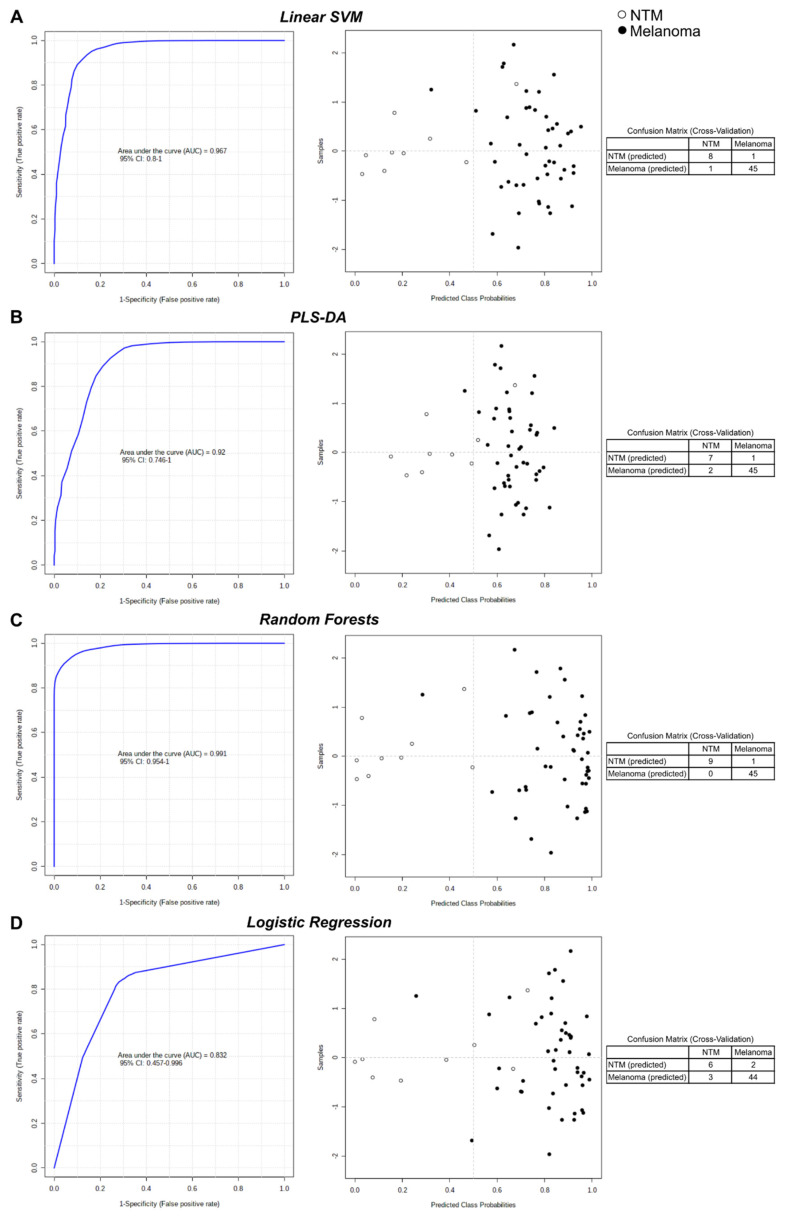
Evaluation of combined biomarker models. Biomarker models based on seven identified biomarker candidates (beta-alanine, *p*-cresol sulfate, sarcosine, tiglylcarnitine, Hex2Cer (d18:1/16:0), Hex2Cer (d18:1/20:0), and PC ae C42:4) were created using different algorithms: (**A**) linear support vector machine (SVM), (**B**) PLS-DA, (**C**) random forests, and (**D**) logistic regression. Receiver operating characteristic (ROC) curves for each combined biomarker model are shown on the left; 100 cross-validations were performed and the results were averaged to generate the ROC curves. Averages of predicted class probabilities of each sample in the 100 cross-validations are summarized in the prediction overviews on the right; corresponding confusion matrices are provided next to each prediction overview.

**Table 1 cancers-13-00434-t001:** Plasma biomarker candidates for melanoma. Potential biomarker candidates were selected according to false discovery rate (FDR)-corrected *p*-value < 0.05 and fold change > 1.5. Biomarker performance was evaluated using the area under the receiver operating characteristic curve (AUROC), sensitivity (true positive rate), and specificity (true negative rate) values.

Compound	AUROC	95% CI	Sensitivity	Specificity	*p*-Value	Fold Change (Melanoma/NTM)
Tiglylcarnitine (C5:1)	0.99	0.966–1	89.13%	100.00%	1.58 × 10^−11^	2.88
beta-Alanine	0.97	0.93–1	91.30%	100.00%	1.76 × 10^−7^	1.80
PC ae C42:4	0.95	0.886–0.993	80.43%	100.00%	9.14 × 10^−7^	1.70
Sarcosine	0.93	0.805–1	86.96%	88.89%	4.46 × 10^−5^	1.93
Hex2Cer (d18:1/16:0)	0.89	0.694–0.995	91.30%	88.89%	5.24 × 10^−4^	2.33
Hex2Cer (d18:1/20:0)	0.85	0.657–0.979	86.96%	88.89%	1.97 × 10^−3^	2.08
*p*-Cresol sulfate	0.84	0.694–0.995	97.83%	77.78%	6.72 × 10^−7^	3.50

## Data Availability

Data are available at https://doi.org/10.5281/zenodo.4457420.

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
