# Peer review of "Targeted Metabolomics Identifies Plasma Biomarkers in Mice with Metabolically Heterogeneous Melanoma Xenografts"

_cancers, 2021, doi:10.3390/cancers13030434_

Round 1
Reviewer 1 Report
This manuscript provides extensive metabolomic data derived from tumors and plasma of mice injected with human melanoma cell lines A375 (BRAF mutant), WM47 (BRAF mutant), WM3000 (NRAS mutant), and WM3311 (triple-wildtype) compared to healthy control. Differences in amino acid as well as lipid metabolites were observed between melanoma subtypes which are, remarkably, mostly irrespective of the genetic driver mutation. Some of the metabolites might serve as biomarkers in melanoma patients. The wealth of data and the pathway analyses performed makes this manuscript a valuable addition for the planned special issue.
However, two major points need to be addressed:
- There is no information on the tumor volume. This is important for comparison between the model systems and especially with respect to a potential use as biomarkers in humans. What is the (estimated) minimal tumor size needed for the use as biomarkers in humans?
- The differences between the tumors derived from the four cell lines are remarkable. The authors speculate (understandably) that this might be due other mutations present. Therefore, extensive information the mutational profile of the cell lines used should be provided and discussed.
Minor:
- The information presented on the controls animals just states “tumor free”. Were they mock injected? Mock injection using Matrigel would be preferable.
Reviewer 2 Report
The authors have used a targeted metabolomics approach to identify melanoma biomarkers in the plasma by utilizing the mice xenograft models. The study holds important clinical relevance if the mice study results can be reproduced in human melanoma patients.
I have a few questions and suggestions for the authors.
- Simple Summary, page 1, line23: Replace "human melanomas" with "human melanoma cell lines." The mice were injected with melanoma cell lines and not with human melanoma tissues.
- Abstract, Page 1, line 35: Indicate the three wild type genes in the triple wildtype cell line for clarity.
- Introduction, Page 2: The authors have discussed the "Warburg effect" that has been considered very important in the context of cancer cell metabolism. However, over the years, the "reverse Warburg effect" has gained a lot of attention. This effect might be responsible for high OXPHOS activity in some melanomas and could be discussed in the introduction section.
- The study did not include the NF1 mutant cell line. Since most of the NF1 mutations are loss of function mutations in melanomas, an NF1 null melanoma cell line could have also served the purpose. Is there any reason for not including the NF1 mutant/null melanoma cell line in the study?
- The authors have not included any details related to the tumors' size at the time of collection? Did all the cell lines grow at a similar rate in the mice? The size of the tumors is important in determining the extent of necrosis. Large tumors tend to have more necrotic tissue at the center of the tumors. This could be the reason for observing the difference in the level of tumor necrosis among the four different melanoma xenografts.
